# Self-Cleaning Highly Porous TiO_2_ Coating Designed by Swelling-Assisted Sequential Infiltration Synthesis (SIS) of a Block Copolymer Template

**DOI:** 10.3390/polym16030308

**Published:** 2024-01-23

**Authors:** Khalil D. Omotosho, Vasanta Gurung, Progna Banerjee, Elena V. Shevchenko, Diana Berman

**Affiliations:** 1Materials Science and Engineering Department, University of North Texas, 1155 Union Circle, Denton, TX 76203, USA; khalildolapoomotosho@my.unt.edu (K.D.O.); vasantagurung@my.unt.edu (V.G.); 2Center for Nanoscale Materials, Argonne National Laboratory, Argonne, IL 60439, USA; pbanerjee@anl.gov; 3Department of Chemistry, James Franck Institute, University of Chicago, Chicago, IL 60637, USA

**Keywords:** porous metal oxides, titanium oxide, coatings, self-cleaning, superhydrophilicity, methylene blue

## Abstract

Photocatalytic self-cleaning coatings with a high surface area are important for a wide range of applications, including optical coatings, solar panels, mirrors, etc. Here, we designed a highly porous TiO_2_ coating with photoinduced self-cleaning characteristics and very high hydrophilicity. This was achieved using the swelling-assisted sequential infiltration synthesis (SIS) of a block copolymer (BCP) template, which was followed by polymer removal via oxidative thermal annealing. The quartz crystal microbalance (QCM) was employed to optimize the infiltration process by estimating the mass of material infiltrated into the polymer template as a function of the number of SIS cycles. This adopted swelling-assisted SIS approach resulted in a smooth uniform TiO_2_ film with an interconnected network of pores. The synthesized film exhibited good crystallinity in the anatase phase. The resulting nanoporous TiO_2_ coatings were tested for their functional characteristics. Exposure to UV irradiation for 1 h induced an improvement in the hydrophilicity of coatings with wetting angle reducing to unmeasurable values upon contact with water droplets. Furthermore, their self-cleaning characteristics were tested by measuring the photocatalytic degradation of methylene blue (MB). The synthesized porous TiO_2_ nanostructures displayed promising photocatalytic activity, demonstrating the degradation of approximately 92% of MB after 180 min under ultraviolet (UV) light irradiation. Thus, the level of performance was comparable to the photoactivity of commercial anatase TiO_2_ nanoparticles of the same quantity. Our results highlight a new robust approach for designing hydrophilic self-cleaning coatings with controlled porosity and composition.

## 1. Introduction

Titanium dioxide (TiO_2_) is considered a favorable candidate for various applications among other semiconductors due to its superior optoelectronic properties, photostability, abundance in nature, structural stability, non-toxicity, and low cost [1,2,3]. In addition to all these characteristics, titanium dioxide is highly attractive for its self-cleaning ability that can be utilized in optical coating, roof tiles, car mirrors, and solar panels [4,5,6,7,8] as anti-reflective coatings, anti-bacterial surfaces [9,10,11,12], self-cleaning fabrics [13,14,15], and blockers of ultra-violent light [16,17,18]. The self-cleaning property arises as a result of the photocatalytic decomposition of organic contaminants [1,3,19] in combination with superhydrophilicity, which enables water droplets to spread on surfaces, thereby washing away degraded contaminants [4,20]. Previous work indicated that among three crystalline polymorphs of TiO_2_, brookite, anatase, and rutile phases, the anatase phase is the most photocatalytically active [21].

The self-cleaning characteristics of titanium oxide depend on the surface access to this photocatalyst with a larger surface-to-volume ratio showing improved properties [1]. Therefore, efforts are dedicated toward the design of highly porous or nanostructured titanium oxide materials. Some of the techniques aimed at achieving this include hydrothermal [22,23,24], sol–gel [25,26,27], and template-assisted [28] and anodization [29]. Some of these techniques enable the synthesis of TiO_2_ nanostructures, in the form of nanoparticles or nanorods, dispersed in solvents that can be further used in the fabrication of conformal coatings. Others form porous films during synthesis; however, the control of the porosity is very limited. Moreover, the process is usually time consuming and incompatible with various substrate materials.

Sequential infiltration synthesis (SIS) has been previously shown as a site-selective technique for the infiltration of inorganic precursors in polymer templates to create organic–inorganic hybrid composite structures and final all-inorganic structures [30,31,32,33,34,35,36]. The SIS relies on diffusion-controlled penetration and the subsequent chemisorption of inorganic molecules inside a polymer template, thus allowing for effective control of the structure and composition of the resulting materials. In order to create all-inorganic porous structures, the removal of the polymer templates can be accomplished via thermal annealing, UV ozone, or oxygen plasma treatment slightly affecting the crystallinity of the films [37,38].

Notably, the majority of the SIS research focuses on alumina infiltration using a trimethylaluminum (TMA) precursor. This is due to the fact that other precursors, such as titanium tetrachloride for TiO_2_ and diethyl zinc for ZnO synthesis, interact weakly with block copolymers (BCP); hence, they are characterized by slower diffusion and a lower adsorption of the metal oxide complexes [39]. Therefore, in the prior effort, the priming of the polymers with TMA has been suggested as a strategy to selectively modify BCP domains via first exposing them to TMA and H_2_O to provide reactive sites for the nucleation of other SIS materials which facilitates their growth within the polymers [30].

Previously, we have shown that by introducing the swelling step which involves soaking the BCP in ethanol at 75 °C for 1 h, we can successfully synthesize porous conformal ZnO coatings [40] without the need for priming, which would alter the chemistry of the resulting structure created. Swelling in ethanol, being selective to polar domains of the BCP, introduces additional reactive sites and porosity in the template, leading to more efficient infiltration. For example, it was demonstrated that the swelling of BCP in ethanol resulted in a four-fold increase in porosity volume, which in turn increased the infiltration depth of the precursor and enabled the synthesis of nanoporous alumina films and structures of high thickness and large volume [37]. Therefore, this swelling step not only gave flexibility in terms of material selection for the growth of inorganic materials but also allowed the creation of larger quantities of materials. The swelling-assisted SIS has been utilized in creating highly porous heterostructures for catalysis [41] and for mechanically robust porous alumina anti-reflective coatings [42,43]. Recently, we showed that the porous AlO_x_ and ZnO coatings synthesized via the swelling-assisted SIS can withstand various environmental conditions such as temperature fluctuations, humidity, vibrations, and physical impacts without compromising their properties, which makes them suitable for various application systems [44].

In this work, we report the synthesis of uniform highly porous TiO_2_ conformal coating by the swelling-assisted sequential infiltration synthesis (SIS) of a block copolymer template (BCP) (Polystyrene 75-b-Polyvinylpyridine 25). The novelty of this work lies in the facile synthesis of the porous TiO_2_ conformal coatings. Importantly, the process does not require priming of the block copolymer template with TMA. They show that very high hydrophilicity can be photoinduced. The self-cleaning properties are demonstrated using methylene blue degradation as a model system. Our results demonstrate a robust approach for designing highly hydrophilic self-cleaning coatings with controlled porosity and composition.

## 2. Materials and Methods

### 2.1. Material Synthesis

We used a block copolymer (BCP) polystyrene-block-poly-4-vinyl pyridine (PS75-b-P4VP25) with number-average molecular weights (Mn) of 75,000 for the polystyrene domain and 25,000 for the poly-4-vinyl pyridine domain with a Polydispersity Index (PDI) of 1.09. The BCP was purchased from Polymer Source Inc. (Dorval, Canada) and dissolved in toluene to make a polymer solution of 20 mg/mL concentration. The polymer solution was filtered through 0.4 µm pore size poly(tetrafluoroethylene) syringe filters (Fisher Scientific, Hampton, VA, USA) to remove agglomeration and non-dissolved polymer powders, which was followed by spin coating at 1500 rpm and 1000 rpm on the surfaces of AT-cut (oscillating in a shear mode) QCMs and ultrasonically cleaned silicon wafers, respectively, for 50 s. For adhesion of the polymer film to the substrates, the polymer was baked at 70 °C for 30 min. To improve the efficiency of polymer infiltration and to enable the creation of thicker inorganic coatings, the polymer was swelled by immersion in a beaker filled with ethanol on a hot plate at a temperature of 75 °C for 1 h. Thereafter, the swelled polymer was allowed to dry in the fume hood for 3 h to remove entrapped ethanol molecules.

The sequential infiltration synthesis (SIS) of the BCP with TiO_2_ was carried out in a Veeco Savannah S100 ALD (Plainview, NY, USA) system with a reactor chamber operated at 90 °C with a precursor temperature of 70 °C and a base pressure of 430 mTorr. The nitrogen flow of 20 sccm was introduced into the reactor chamber as the carrier gas for the precursors. Following this, the BCP samples were exposed to 20 cycles of titanium tetrachloride (TiCl_4_) metal–organic precursor from Sigma Aldrich (St. Louis, MO, USA) with 2 s pulsing for each cycle and a total exposure time of 600 s, after which the excess reactant was purged with 100 sccm of compressed dry air (CDA) for 20 s. DI water pulsed at 4 s for 20 cycles was further introduced into the chamber to act as a reactant with the TiCl_4_ with a total exposure time of 400 s, which was followed by purging with CDA (100 sccm) for 300 s to remove excess by-products.

After infiltration of the polymer with TiO_2_ precursors, the polymer template was removed via oxidative thermal annealing in a Thermo Scientific Lindberg Blue M Furnace (Hampton, VA, USA) for 4 h at 450 °C under airflow.

### 2.2. QCM Analysis

Meanwhile, for the QCM studies, the polymer templates were removed by UV ozone using UVOCS T16x16 OES (Lansdale, PA, USA) with a 254 nm UV wavelength at room temperature due to the incompatibility of the QCM crystals with high-temperature processing. The resulting thickness of the TiO_2_ film after the polymer removal is approximately 500 nm ± 15. The Quartz Crystal Microbalance (QCM) proves to be a highly responsive method for nonintrusive quantitative assessment of deposited material quantities. In this study, we employed gold-coated QCM crystals of 1 inch in diameter with a base resonant frequency of 5 MHz, which were obtained from Inficon (Bad Ragaz, Switzerland). To monitor the resonant frequency, an SRS QCM200 controller from Stanford Research Systems (Sunnyvale, CA, USA) was utilized.

In the classic approach, the change in frequency of the QCM, ∆*f*, under applied load, caused by the deposited mass, *m*, is [45]:(1)$$∆f=-\frac{{2f}^{2}}{A\sqrt{\rho_{q}{µ}_{q}}}∆m$$
where *f*_o_ is the fundamental frequency of the QCM, $\rho_{q}$ is the density of quartz (2.648 g/cm^3^), ${µ}_{q}$ is the shear modulus of quartz (2.947 × 10^11^ g cm^−1^ S^−2^), and *A* is the QCM surface area.

### 2.3. Characterization

A FEI Quanta 200 ESEM (Hillsboro, OR, USA) microscope equipped with an EDX EDS system was utilized for the scanning electron microscopy (SEM) imaging of the coatings and elemental composition. Transmission Electron Microscopy (TEM) was conducted with the JEOL 2100F instrument (Tokyo, Japan). X-ray Photoelectron Spectroscopy (XPS) chemical analysis was carried out using the PHI 5000 Versaprobe (Chanhassen, MN, USA) scanning XPS spectrometer with monochromatic 1486.6 eV Al Kα radiation. All binding energies were corrected for the charge shift using the C 1s peak at 284.6 eV. X-Ray Diffraction (XRD) analysis was performed with Bruker D2 Phaser (Billerica, MA, USA) and Bruker D8 Discover. Changes in the vibrational spectrum of the polymer after infiltration were monitored with the Thermo Scientific (Hampton, VA, USA) Nicolet 6700 Fourier transform infrared spectrometer (FTIR) with a 1000–4000 cm^−1^ spectral range equipped with a variable angle grazing angle attenuated total reflection (GATR-ATR) accessory (Vari-GATR, Harrick Scientific, Pleasantville, NJ, USA). Contact angle measurements were obtained by the Sessile water drop method using a Dataphysics OCA 15EC (Riverside, CA, USA) contact angle goniometer equipped with a CCD video camera. A J.A. Woollam (Lincoln, NE, USA) horizontal M-2000 ellipsometry system was used for the porosity estimation by fitting the model created to the generated data using Bruggeman’s model.

### 2.4. Photocatalytic Activity

The evaluation of the photocatalytic degradation of methylene blue (MB) was performed by exposure of the TiO_2_ photocatalyst to short wavelength ultraviolet light (UV, 254 nm) at 30-min intervals. The TiO_2_ catalyst was synthesized by soaking a paper filter in a thick BCP polymer solution, which was followed by swelling and infiltration. The polymer and paper filter were removed by burning them off in the furnace at 450 °C for 4 h, leaving a remnant of TiO_2_ flakes. The anatase phase of titanium(IV)oxide was purchased from Sigma Aldrich for comparison of the photocatalytic activity of our synthesized TiO_2_ with a commercial anatase TiO_2_ (<25 nm in size). The MB powder purchased from Sigma Aldrich was diluted with DI water to make a concentration of 0.01 g/L. For consistency, 5 mg of both the commercial TiO_2_ and the synthesized TiO_2_ powders were added to a 100 mL MB solution to make 2 separate solutions. The suspension was sonicated and then stirred in the dark before irradiation in a Cole–Parmer UV viewing cabinet equipped with a 15-watt shortwave (254 nm) ultraviolet tube under constant stirring at 200 rpm at 30 min intervals up to 180 min. A Shimadzu (Kyoto, Japan) UV-1800 UV-Vis spectrophotometer was used to measure the absorbance spectra. The concentration of MB degraded at each time interval was calculated using the Beer–Lambert law, which describes the direct relationship between the amount of absorbed light by the solution and its concentration with this equation:(2)A=εlc
where *A* is the absorbance, ε is the extinction coefficient, *l* is the optical path length of the quartz cuvette, and *c* is the concentration of the solution.
(3)$$\%\mathrm{Degradation}\;\mathrm{of}\;\mathrm{MB}\;=\frac{Ci-Ct}{Ci}$$
where *C_i_* is the initial concentration of MB in the dark and *C_t_* is the concentration of MB after UV irradiation time, *t*.

## 3. Results and Discussion

### 3.1. Mass Change and FTIR

The nanoporous titanium oxide films have been synthesized using the swelling-modified SIS approach. The initial BCP template was swollen in ethanol, which was followed by infiltration with titanium oxide precursors and then by the thermally assisted removal of the polymer template, which resulted in the deposition of a porous titanium oxide layer (Figure 1).

The optimization of the infiltration effectiveness has been performed using QCM analysis (Figure 2). For this, the mass of the polymer template exposed to swelling and infiltration of the TiO_2_ precursors was calculated from the changes in the resonant frequency of the QCM after each processing step (Figure 2a). There was no change in mass observed after swelling of the BCP template in ethanol, indicating no dissolving or delamination of the BCP template from the substrate upon immersion in ethanol. The major effect of the swelling step is the introduction of additional porosity due to the selective opening of the micelles as well as an increase in the number of polar groups acting later as the infiltration sites [37].

A sharp increase in mass was observed after two cycles of TiO_2_ infiltration from ~32 to 70 µg/cm^2^. A further increase in the number of infiltration cycles resulted in a steep rise in the mass change. Full infiltration of the swollen BCP was reached after 20 cycles, resulting in a total mass of 225 µg/cm^2^. Prior to the first infiltration cycle, there were sufficient functional groups in pores available for molecular attachment and diffusion, respectively, hence the significant mass change. With an increasing number of infiltration cycles, the functional groups became less available, and the diffusion coefficient dropped significantly.

FTIR absorption spectra (Figure 2b) were measured for the block copolymer, swollen polymer, and the swollen polymer infiltrated with TiO_2_ in the ATR mode over the range of 650–3600 cm^−1^. The peaks at 1450 cm^−1^, 1493 cm^−1^, and 1600 cm^−1^ correspond to CH_2_ bending and C=C stretching of the aromatic rings in the polystyrene block (PS) [46]. The peaks at 822 cm^−1^, 993 cm^−1^, 1415 cm^−1^, 1557 cm^−1^, and 2969 cm^−1^ are related to the poly(4-vinylpyridine) block (P4VP). These peaks are assigned to C-H out-of-plane bending vibration, C-H in-plane bending vibration, CH bending, C=N stretching and C-H stretching, respectively [47,48,49]. The broad band at 2361 cm^−1^ is due to the presence of CO_2_ in the instrument. It is evident that after the SIS step, the BCP-assigned peaks were significantly diminished.

### 3.2. Structure and Surface Morphology

The synthesized films, after the thermally assisted polymer removal step, were analyzed for their structure and composition (Figure 3). The characteristic peaks associated with crystalline anatase TiO_2_ can be identified in the XRD pattern (Figure 3a) for the synthesized porous TiO_2_ film. The notably high intense peak at 2θ = 25° corresponds to the (101) diffraction plane of the anatase phase, which is the preferred orientation [50,51,52]. Among the polymorphs of TiO_2_, the anatase phase is the most favorable polymorph due to its superior photocatalytic activity compared to the rutile and brookite TiO_2_ phases [53]. Anatase has a higher band gap (~3.2 eV) vs 3.0 for rutile, resulting in raising the valence band maximum to higher energy levels relative to redox potentials of adsorbed molecules, thereby facilitating electron transfer from TiO_2_ to adsorbed molecules [53].

The XRD data (Figure 3a) suggest that the annealing temperature of 450 °C is sufficient to synthesize a highly crystalline TiO_2_ film. Figure 3b shows the low-magnification and high-magnification high-resolution TEM images of the porous TiO_2_ coating. The lattice fringe spacing between two adjacent crystal planes of the particles in the TEM image was estimated to be 0.347 nm, corresponding to the (101) lattice plane of anatase [54,55]. Moreover, the low-magnification TEM image reveals a network of interconnected TiO_2_ nanotubes.

The surface morphology of the porous TiO_2_ coating was analyzed with scanning electron microscopy (SEM). Figure 3 depicts the low magnification (Figure 3c) and high magnification (Figure 3d) images of the porous coating. We see that the synthesized porous TiO_2_ coating surface is uniformly smooth, revealing a network of interconnected porous TiO_2_ tubes with varying size distribution at higher magnification. From the ellipsometry measurement, the porosity of the coating was estimated to be ~80 ± 5%. The EDS analysis confirmed the presence of Ti in the coating.

### 3.3. Wetting Properties and Surface Composition

Figure 4a summarizes the effect of UV exposure on the wetting properties of the porous TiO_2_ coating. Prior to UV exposure, the measured contact angle for the porous coating was 25 ± 1°, indicating that the coating was hydrophilic with high affinity to water. However, after exposure to UV light for 1 h, we observed a dramatic decrease in the contact angle to about 5 ± 0.4°, which suggests that the exposure resulted in a superior hydrophilicity of the coating [56,57]. Upon further exposure to UV light, no change in the contact angle was observed; hence, the very high hydrophilicity property of the TiO_2_ surface was maintained. Highly hydrophilic surfaces are known to support the spread of water, which favors easy removal of surface contaminants.

The XPS spectra in Figure 4b–d reveal the surface chemical composition of the porous TiO_2_ coating before (Figure 4b,c) and after UV irradiation (Figure 4d,e) to investigate the mechanism of hydrophilicity of the TO_2_ coating upon UV exposure. The Ti 2p_3/2_ XPS spectra in (Figure 4b,d) were fitted to a single Gaussian peak with a binding energy of 458.5 eV corresponding to stoichiometric TiO_2_ in the lattice with a +4 oxidation state for Ti [58]. No changes in peak binding energy were observed for the Ti 2p_3/2_ spectra before and after UV irradiation. Figure 4c,e show the O 1s XPS spectra with each spectrum deconvoluted into two sub-peaks with a Gaussian distribution. The peaks with identical binding energies of 529.8 eV can be attributed to lattice oxygen in TiO_2_. However, it is worthy of note that there is a binding energy shift of 0.5 eV: from 531.6 eV (Figure 4c) before UV exposure to 532.1 eV (Figure 4e) after UV exposure. These peaks are assigned to hydroxyl groups (Ti-OH). The significant shift in binding energy could be a result of oxygen vacancies created in the coating upon UV exposure. Furthermore, we noticed an increase in the amount of surface OH groups of the TiO_2_ film after UV exposure, which was evidenced by a 12% increase in the area of the OH group peaks [59,60]. We attribute the increased presence of OH groups on the UV-irradiated film to the oxygen vacancies created. It can be deduced from the chemical analysis that the photoinduced increase in the hydrophilicity of the TiO_2_ film is due to the increased surface OH groups that tend to have a high affinity for water.

### 3.4. Photocatalytic Degradation of Methylene Blue

To demonstrate the photocatalytic degradation performance of the synthesized TiO_2_ coating, we measured the changes in the UV-VIS absorption spectra using the UV-vis spectrophotometer, as shown in Figure 5. Also, the performance of our synthesized TiO_2_ was compared with a commercial anatase TiO_2_ powder. The snapshots of the MB solution after each exposure to UV light with and without the addition of the TiO_2_ catalyst were captured to observe the change in coloration or lack thereof. For decades, TiO_2_ has been widely used as an ideal photocatalyst for the destruction of organic contaminants in water due to its non-toxic nature, stability in aqueous suspension, and insolubility in water [61]. Matthews et al. [62] and Reeves et al. [63] were among the earliest researchers to demonstrate the photocatalytic degradation of MB with TiO_2_, since MB dye is widely used as a model system of pollutants. It is worth noting that when TiO_2_ is irradiated with an energy source greater than its band gap, typically ≥3.2 eV, photon absorption occurs, leading to electrons excitation from the valence band to the conduction band, thereby generating exciton pairs, leaving the holes in the valence band [4,64]. The electrons and holes can undergo either radiative recombination or non-radiative recombination at the surface or within the bulk of the semiconductor due to crystal imperfections [20]. Excitons that do not undergo recombination tend to react with adsorbed dye molecules undergoing oxidation and reduction half-cycles, resulting in the decomposition of volatile organic compounds. Electrons in the conduction band react with an acceptor, usually surface oxygen molecules, and reduce it into superoxide radicals which can react with water to form hydroxyl radicals. Meanwhile, the holes in the valence band react with water or the surface hydroxyl group to form hydroxyl radicals. These radical species aid the decomposition of volatile organic compounds into CO_2_ and H_2_O by oxidation of the organic molecules [20,65]. For effective performance of the TiO_2_ for photocatalysis, the charge recombination or charge annihilation needs to be minimized.

Figure 5a shows the UV-vis absorbance spectra of the MB solution recorded in the spectrum range of 400–800 nm. In the absence of TiO_2_, the maximum value of the absorbance peak slowly decreases with UV irradiation time, indicating that the MB degradation rate is low. The lack of discoloration of the MB even after 180 min of UV exposure supports this claim. However, after adding the TiO_2_ catalysts Figure 5b,c, the absorbance peak disappears rapidly and becomes nearly flat after UV irradiation for 180 min, while the MB solution becomes colorless, indicating complete degradation. The performances of the SIS-synthesized TiO_2_ are very comparable with the commercial anatase TiO_2_ powder.

To investigate the stability and the reusability of our synthesized TiO_2_ photocatalyst, we analyzed the photodegradation performance of MB solution with the previously used SIS synthesized TiO_2_ powders, as shown in Figure 6a. The absorption spectra demonstrate the effective degradation of MB solution in the presence of the used TiO_2_ powders, which was evidenced by the disappearance of the peak and discoloration. This observation suggests that the SIS-synthesized TiO_2_ coatings are stable over time and can be reused effectively.

Figure 6b presents a summary of the photocatalytic degradation activity of MB. After irradiation for 30 min, over 50% of the MB solution had already been degraded in the presence of the pure TiO_2_. In comparison to the performance of the SIS-synthesized TiO_2_, 40% of the MB was degraded after 30 min. When the SIS-synthesized TiO_2_ was reused, about 30% of the MB solution degraded after 30 min of UV irradiation time. After 180 min, approximately 40% of the MB solution was decomposed in the absence of TiO_2_. However, with the addition of the TiO_2_ catalysts, the MB degradation rate was significantly enhanced from 40% to ~92% and 96% for the SIS TiO_2_ and pure TiO_2_, respectively, suggesting excellent photocatalytic degradation performance of the synthesized TiO_2_. Interestingly, approximately 80% of the MB solution was decomposed with the previously used SIS-synthesized TiO_2_ following 180 min of UV irradiation, demonstrating the efficiency and stability of our TiO_2_ coating.

## 4. Conclusions

In this study, we successfully synthesized a highly porous TiO_2_ coating that exhibits efficient photocatalytic self-cleaning activity. Our approach relies on the swelling-assisted sequential infiltration synthesis (SIS) of a block copolymer (BCP) template eliminating the need for priming the polymer with TMA. Subsequent thermally assisted polymer removal results in a TiO_2_ structure replicating the BCP template. This process enables the creation of a uniformly smooth TiO_2_ coating with a network of interconnected pores, showcasing good crystallinity and anatase phase formation. The surfaces of the resulting coating demonstrated high hydrophilicity, which improved even more upon exposure to UV irradiation. Through this synthesis technique, our results reveal excellent photocatalytic degradation activity and stability of the synthesized TiO_2_ catalyst within 180 min. This is evidenced by the discoloration of the solution of a model pollutant, methylene blue (MB), which is accompanied by a flattening of the absorbance peak. The photocatalytic activity of porous TiO_2_ is comparable to the photoactivity of commercial anatase TiO_2_ nanoparticles of the same quantity. Therefore, our approach offers a direct pathway for the design of conformal self-cleaning coatings that can also be integrated into other materials.

## Figures and Tables

**Figure 1 polymers-16-00308-f001:**
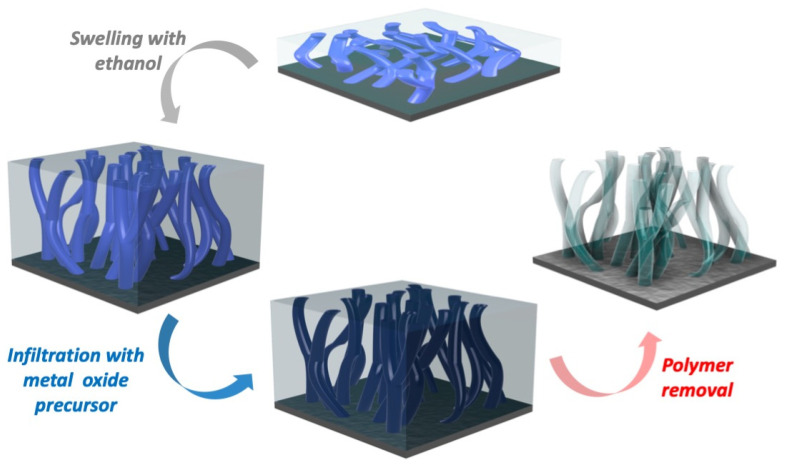
Schematic of the synthesis of porous TiO_2_ via SIS of BCP.

**Figure 2 polymers-16-00308-f002:**
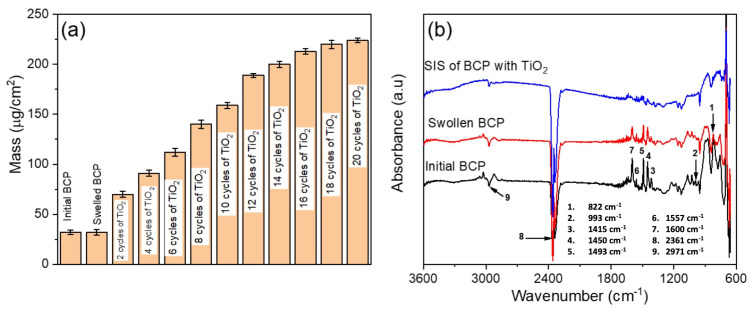
(**a**) Summary of mass change in TiO_2_ film as a function of number of cycles and (**b**) FTIR spectra monitoring stages of the BCP infiltration.

**Figure 3 polymers-16-00308-f003:**
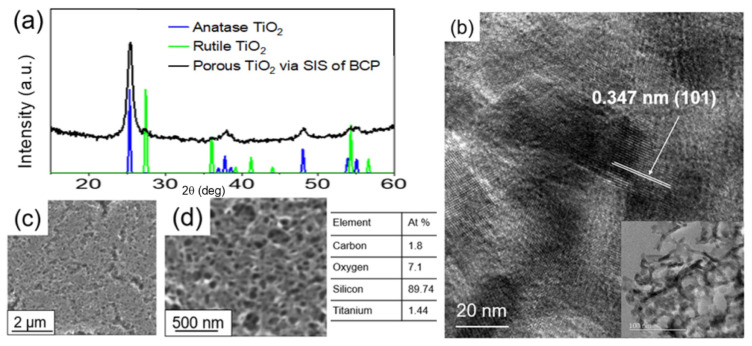
(**a**) XRD spectra and (**b**) high-resolution TEM image (inset: low magnification TEM image). SEM images (**c**,**d**) of the porous TiO_2_ coating with the elemental composition.

**Figure 4 polymers-16-00308-f004:**
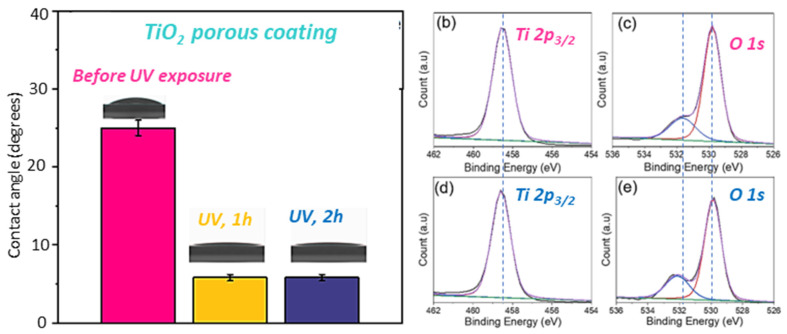
(**a**) Water contact angle dependence on the UV exposure time. XPS spectra of the porous TiO_2_ coatings before (**b**,**c**) and after 2 h UV irradiation (**d**,**e**).

**Figure 5 polymers-16-00308-f005:**
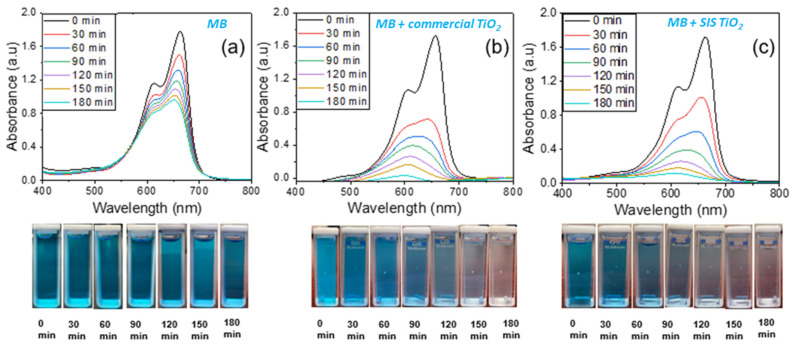
Photocatalytic degradation test at 30-min intervals under UV (254 nm) light exposure (**a**) MB solution (**b**) MB with pure TiO_2_ and (**c**) MB with SIS synthesized TiO_2_.

**Figure 6 polymers-16-00308-f006:**
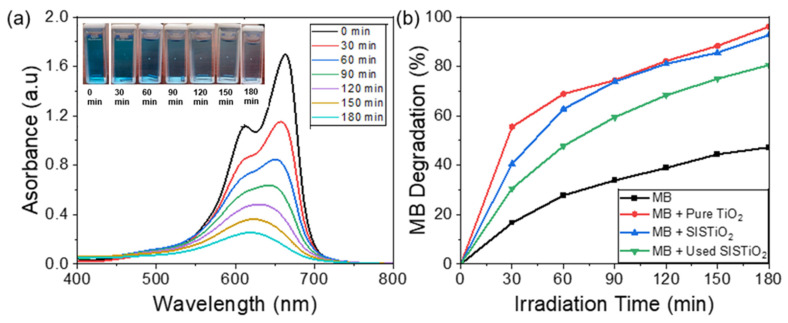
(**a**) Photocatalytic degradation test under UV (254 nm) light for used SIS synthesized TiO_2_. (**b**) Summary of MB photocatalytic degradation activities.

## Data Availability

The authors confirm that the data supporting the findings of this study are available within the article.

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
