# Peer review of "Self-Cleaning Highly Porous TiO2 Coating Designed by Swelling-Assisted Sequential Infiltration Synthesis (SIS) of a Block Copolymer Template"

_polymers, 2024, doi:10.3390/polym16030308_

Round 1
Reviewer 1 Report
Comments and Suggestions for Authors
But there are a number of comments and questions on this work.
Is the used Polystyrene 75-b-Polyvinylpyridine 25 the best choice as a polymer template? What are its advantages? Is it possible to use other polymers? To what extent does the proposed procedure allow the removal of polymer residues from the synthesized material? Is an anatase the only form of titanium oxide in the material? Could there be other highly dispersed amorphous states that are not visible in XRD spectra? What structural changes are taking place upon exposure to UV irradiation when the surface becomes superhydrophilic? Is it possible to observe the conversion of Ti4+ sites to Ti3+ sites? A number of conclusions made by the authors require more rigorous quantitative confirmations. So the authors conclude that they synthesized a highly porous TiO2 coating that demonstrates efficient photocatalytic self-cleaning activity. First of all, what can the authors say about the volume and pore diameter of the resulting material, about its specific surface area? Can these quantities be estimated? The excellent photocatalytic degradation activity of synthesized coating can be discussed only in comparison with other materials based on TiO2, and not in comparison with the results obtained in the TiO2 absence.
What can be said about stability of the produced coatings? Is it possible to reuse them?
In general, it would be useful to compare the characteristics of the original coatings with those after UV irradiation and their application in the photocatalytic decomposition of methylene blue.
The article may be published after major revision.

Author Response
The presented material deals with the design of TiO2 coatings with many potential applications based on their optical and electrophysical properties. One possible use of these materials is the photocatalytic decomposition of organic pollutants. To demonstrate the photocatalytic degradation performance of the synthesized TiO2 coating methylene blue was taken as a model compound. The interesting and original method production of such coatings by the swelling assisted sequential infiltration synthesis (SIS) of a block copolymer template (BCP) (Polystyrene 75-b-Polyvinylpyridine 25) is developed in this research. It can be useful in the design of new efficient supports and catalysts for many ecologically important processes of pollutant oxidation. But there are a number of comments and questions on this work.
Response: We appreciate the reviewer’s comments highlighting the originality and importance of this work.
- Is the used Polystyrene 75-b-Polyvinylpyridine 25 the best choice as a polymer template? What are its advantages? Is it possible to use other polymers?
Response: We appreciate the reviewer’s comment. In our prior work, we explore alternative choices for the polymer templates. For example, we used PIM-1 as a template to design mechanically stable porous all-inorganic antireflective coatings (ACS Nano 2022, 16, 14754-14764). Nevertheless, previous work by our group and others demonstrated that the main advantage of the block copolymer (in comparison to other polymers or PIM-1) is that it enables the design of different patterns of inorganic coatings with different percentage porosity by varying the ratio of the polar to non-polar domains. Here, having higher concentration of non-polar PS, as in case of PS75-b-P4VP25, enables synthesis the creation of highly porous metal oxides with porosity reaching up to 80% depending on the infiltration cycles. At the same time, in comparison to other polymers, the PS75-b-P4VP25 is stable during the swelling step without any signs delamination.
- To what extent does the proposed procedure allow the removal of polymer residues from the synthesized material?
Response: We thank the reviewer for the comment. In our earlier work (ACS Nano 2017, 11, 2521-2530), we demonstrated that the thermal annealing-based polymer removal procedure is very effective, as evidenced by the amount of carbon present in EDX analysis (about 1.8 at%). We attribute the source of this carbon to be adventitious carbon from atmospheric contamination.
- Is an anatase the only form of titanium oxide in the material? Could there be other highly dispersed amorphous states that are not visible in XRD spectra? What structural changes are taking place upon exposure to UV irradiation when the surface becomes superhydrophilic? Is it possible to observe the conversion of Ti4+ sites to Ti3+ sites?
Response: We appreciate the reviewer’s comment. There is a possibility of the existence of other amorphous TiO2 states which are obviously not visible with the XRD. We do not expect to see any structural changes upon exposure to UV irradiation, since the power of the UV lamp is not high enough to induce such a change in the TiO2 coating. We have included a reference to support this claim (Applied Surface Science, 2008, 255(5), 2752-2758). From the XPS analysis of the TiO2 coating before and after UV irradiation, no change in the Ti state was observed, however we do see an increase in OH group concentration from the O1s spectra. We have included this additional discussion in the revised version of the manuscript.
- A number of conclusions made by the authors require more rigorous quantitative confirmations. So the authors conclude that they synthesized a highly porous TiO2 coating that demonstrates efficient photocatalytic self-cleaning activity. First of all, what can the authors say about the volume and pore diameter of the resulting material, about its specific surface area? Can these quantities be estimated?
Response: We understand the reviewer’s concern. In our previous work, we performed in-depth analysis of the porosity evolution during the synthesis steps (ACS Nano 2017, 11 (3), 2521-2530 and Langmuir 2019, 35 (3), 796-803). Similarly, the analysis of the porosity has been performed before by others (Nano Lett. 2012, 12(9), 5033-5038). It should be noted, that the porosity is dictated by the relative concentration of polar-nonpolar domains. Therefore, though we report for the first time the sequential infiltration synthesis (SIS) of titanium oxide, overall structure evolution follows similar trends as previously studied SIS-designed aluminum oxide and zinc oxide.
- The excellent photocatalytic degradation activity of synthesized coating can be discussed only in comparison with other materials based on TiO2, and not in comparison with the results obtained in the TiO2 absence. What can be said about stability of the produced coatings? Is it possible to reuse them? In general, it would be useful to compare the characteristics of the original coatings with those after UV irradiation and their application in the photocatalytic decomposition of methylene blue.
Response: We thank the reviewer for this suggestion. We have conducted an experiment to demonstrate the reusability of our synthesized TiO2 coating and compared the photocatalytic decomposition performance with the original coating. The results are included in the revised version of the manuscript.
Reviewer 2 Report
Comments and Suggestions for Authors
In this manuscript, the author reported the synthesis of porous TiO2 and its photocatalytic property. however, the whole manuscript in bad writing and organizing, major revision was needed.
(1) As mentioned in the title, the obtained TiO2 nanomaterials is porous, and the porous structure is the key factor to the photocatalytic property, thus the BET of TiO2 nanomaterials should provided.
(2) In the Introduction, the novelty of this work should emphasized.
(3) Section 2 and 3 should divided into different subsection, especially for the bold “Photocatalytic Degradation of Methylene Blue”.
(4) Does the “molecular weight” refer to Mn or Mw? And what its PDI?
(5) Why do you introduce QCM analysis? The corresponding results does not provide and discussed.
(6) Figure 1, why does the swollen BCP can form tubular structure? What is the mechanism?
(7) Figure 2a, how can you determined the mass (ug/cm2)? which should mentioned in the Methods.
(8) Figure 2b should discussed in detail, what is the structure for the peak around 2400 cm-1? In addition, the vibration peak for styrene and pyridine should also point out.
(9) Caption of Figure 3e mentioned elemental composition, but where is the EDS?
(10) Figure 4, the contact angle decreased dramatically with UV exposure, what is the mechanism?
(11) Figure 5, the two samples (pure MB and MB with catalyst) at 0 min are not the same colour, which means they are not the same concentration, so the result is not suitable for comparison.
(12) The conclusion should summarize the important result of this work with specific/detail data.
Comments on the Quality of English LanguageModerate editing of English language required
Author Response
In this manuscript, the author reported the synthesis of porous TiO2 and its photocatalytic property. however, the whole manuscript in bad writing and organizing, major revision was needed.
Response: We thank the reviewer for highlighting the crux of this work and for the critique of the manuscript. Below, we provide a detailed response to the reviewer’s comments.
(1) As mentioned in the title, the obtained TiO2 nanomaterials is porous, and the porous structure is the key factor to the photocatalytic property, thus the BET of TiO2 nanomaterials should provided.
Response: We appreciate the reviewer’s comment. Unfortunately, BET is compatible only with the large amounts of material. Meanwhile, we focus on designing nanoporous thin films with the effort to show that they are reliable as self-cleaning surfaces. In our previous work, we performed in-depth analysis of the porosity evolution during the synthesis steps (ACS Nano 2017, 11 (3), 2521-2530 and Langmuir 2019, 35 (3), 796-803). Similarly, the analysis of the porosity has been performed before by others (Nano Lett. 2012, 12(9), 5033-5038). It should be noted, that the porosity is dictated by the relative concentration of polar-nonpolar domains. Therefore, though we report for the first time the sequential infiltration synthesis (SIS) of titanium oxide, overall structure evolution follows similar trends as previously studied SIS-designed aluminum oxide and zinc oxide.
(2) In the Introduction, the novelty of this work should emphasized.
Response: We thank the reviewer for the valuable suggestion. The Introduction section has been revised to emphasize the novelty of the work.
(3) Section 2 and 3 should divided into different subsection, especially for the bold “Photocatalytic Degradation of Methylene Blue”.
Response: We agree with the reviewer. For clarity, sections 2 and 3 have been divided into subsections.
(4) Does the “molecular weight” refer to Mn or Mw? And what its PDI?
Response: The molecular weight refers to Mn and the PDI is 1.09. We have added this information to the experimental section.
(5) Why do you introduce QCM analysis? The corresponding results does not provide and discussed.
Response: We appreciate the reviewer’s comment. QCM is a very sensitive technique to provide insights into the infiltration steps, even at very small quantities of the polymer. Specifically, the use of the QCM allowed to quantitatively evaluate stability of the polymer during the swelling step by monitoring the changes in resonant frequency before and after swelling. Next, the QCM helped to evaluate efficiency of infiltration of the TiO2 precursors in the polymer by estimating the mass of the infiltrated material. To highlight this, we added additional discussion in the revised version of the manuscript.
(6) Figure 1, why does the swollen BCP can form tubular structure? What is the mechanism?
Response: We appreciate the reviewer’s comment. The detailed analysis of the morphological changes during the swelling of the BCP were previously reported by others (for example, Nano Lett. 2012, 12(9), 5033-5038). In short, the swelling of the BCP in ethanol associated with formation of tubular structure is due to the opening of the BCP micelles after swelling in ethanol. Exposure of the BCP to polar solvents, such as ethanol, leads to the opening of the polar micelles (PVP) as a result of their swelling caused by the polar solvent molecules diffused through the nonpolar domains (PS). Swollen PVP domains expand and create pressure in the polymer film. When the polymer is removed from the swelling agent, fast drying of ethanol and a predefined form of the polystyrene part of the polymer prevent relaxation of the film back to the initial structure, and thus residual porosity (of tubular structure) is created. We have added additional discussion in the revised version of the manuscript.
(7) Figure 2a, how can you determined the mass (ug/cm2)? which should mentioned in the Methods.
Response: The mass was calculated from equation (1) under section 2.2., after recording the change in frequency ∆f associated with the deposited mass. We have clarified this in the text.
(8) Figure 2b should discussed in detail, what is the structure for the peak around 2400 cm-1? In addition, the vibration peak for styrene and pyridine should also point out.
Response: We thank the reviewer for this suggestion. We have included this information in the results and discussion section.
(9) Caption of Figure 3e mentioned elemental composition, but where is the EDS?
Response: We thank the reviewer for this comment. This has been corrected in the revised version of the manuscript.
(10) Figure 4, the contact angle decreased dramatically with UV exposure, what is the mechanism?
Response: We appreciate the reviewer’s comment. We have expanded the discussion of the photoinduced superhydrophilic mechanism in the discussion section of the revised manuscript.
(11) Figure 5, the two samples (pure MB and MB with catalyst) at 0 min are not the same colour, which means they are not the same concentration, so the result is not suitable for comparison.
Response: We understand the reviewer’s concern. The concentration was kept the same for both experiments. The difference in color is an effect of the TiO2 addition to the MB solution.
(12) The conclusion should summarize the important result of this work with specific/detail data
Response: We thank the reviewer for this comment. The conclusion section has been revised.
Reviewer 3 Report
Comments and Suggestions for Authors
The authors report self-cleaning highly porous TiO2 coating designed by swelling-assisted sequential infiltration synthesis of a block copolymer template. The results show that treatment of the coatings with UV irradiation for 1 hour led to superhydrophilicity effect upon their contact with water droplets. And their self-cleaning characteristics were tested by measuring photocatalytic degradation of methylene blue. The synthesized TiO2 structures demonstrated approximately 90% MB deg-radation after 180 minutes of ultraviolet light irradiation in contrast to 40% degradation observed without them. The results are very interesting. However, some points of the manuscript should be improved. Specific comments are given below.
1. The pure TiO2 should be used as control example to compare the catalytic ability.
2. The catalytic mechanism of MB with TiO2 catalyst should be measured.
3. The catalytic ability of TiO2 catalyst should be compared with other catalyst.
4. The hydrodynamic size and zeta potential of TiO2 catalyst should be measured.
5. The specific surface area of TiO2 catalyst should be measured.
Comments on the Quality of English LanguageMinor editing of English language required
Author Response
The authors report self-cleaning highly porous TiO2 coating designed by swelling-assisted sequential infiltration synthesis of a block copolymer template. The results show that treatment of the coatings with UV irradiation for 1 hour led to superhydrophilicity effect upon their contact with water droplets. And their self-cleaning characteristics were tested by measuring photocatalytic degradation of methylene blue. The synthesized TiO2 structures demonstrated approximately 90% MB degradation after 180 minutes of ultraviolet light irradiation in contrast to 40% degradation observed without them. The results are very interesting. However, some points of the manuscript should be improved. Specific comments are given below
Response: We thank the reviewer for highlighting the technical importance of this work.
- The pure TiO2 should be used as control example to compare the catalytic ability.
Response: We appreciate the reviewer’s suggestion. We have conducted an experiment to compare the photocatalytic decomposition performance of pure TiO2 with our synthesized TiO2 and the new results have been included in the revised version of the manuscript.
- The catalytic mechanism of MB with TiO2 catalyst should be measured.
Response: We have included additional discussion of the catalytic mechanism for the MB degradation with TiO2 catalyst in the results and discussion section.
- The catalytic ability of TiO2 catalyst should be compared with other catalyst.
Response: We appreciate the reviewer’s comment. As suggested by the reviewer, we have included additional results comparing the performance of the synthesized nanoporous structures to the pure titania catalyst purchased from Sigma Aldrich.
- The hydrodynamic size and zeta potential of TiO2 catalyst should be measured.
Response: We appreciate the reviewer’s comment. We believe that such measurements would be valuable for the future work.
- The specific surface area of TiO2 catalyst should be measured.
Response: We appreciate the reviewer’s comment. Our previously published paper (ACS Appl. Mater. Interfaces 2021, 13, 35941-35948) gives an indication of the surface area of the coatings. Specifically, the prior analysis indicated that the SIS-designed nanoporous metal oxides have ∼80 m2/g of the available surface area. Therefore, though we report for the first time the sequential infiltration synthesis (SIS) of titanium oxide, overall structure evolution follows similar trends as previously studied SIS-designed aluminum oxide.
Round 2
Reviewer 1 Report
Comments and Suggestions for Authors
This answer doesn't look convincing. Some porosity characteristics should be given or at least estimated. "We understand the reviewer’s concern. In our previous work, we performed in-depth analysis of the porosity evolution during the synthesis steps (ACS Nano 2017, 11 (3), 2521-2530 and Langmuir 2019, 35 (3), 796-803). Similarly, the analysis of the porosity has been performed before by others (Nano Lett. 2012, 12(9), 5033-5038). It should be noted, that the porosity is dictated by the relative concentration of polar-nonpolar domains. Therefore, though we report for the first time the sequential infiltration synthesis (SIS) of titanium oxide, overall structure evolution follows similar trends as previously studied SIS-designed aluminum oxide and zinc oxide".
Author Response
Reviewer: 1
This answer doesn't look convincing. Some porosity characteristics should be given or at least estimated. "We understand the reviewer’s concern. In our previous work, we performed in-depth analysis of the porosity evolution during the synthesis steps (ACS Nano 2017, 11 (3), 2521-2530 and Langmuir 2019, 35 (3), 796-803). Similarly, the analysis of the porosity has been performed before by others (Nano Lett. 2012, 12(9), 5033-5038). It should be noted, that the porosity is dictated by the relative concentration of polar-nonpolar domains. Therefore, though we report for the first time the sequential infiltration synthesis (SIS) of titanium oxide, overall structure evolution follows similar trends as previously studied SIS-designed aluminum oxide and zinc oxide".
Response: We appreciate the reviewer’s comments. We have included the porosity analysis in the revised version of the manuscript.
Reviewer 2 Report
Comments and Suggestions for Authors
All of the issues mentioned were resolved in detail
Comments on the Quality of English LanguageModerate editing of English language required
Author Response
We thank the reviewer for careful examination of our manuscript.
Reviewer 3 Report
Comments and Suggestions for Authors
The authors have addressed the problem very well, and the manuscript can be accepted in the present form.
Comments on the Quality of English LanguageMinor editing of English language required
Author Response

(The authors gave the same response as above.)

Round 3
Reviewer 1 Report
Comments and Suggestions for Authors
It's ok now. Can be published.